

# Novel real-time PCR based assays for differentiating fall armyworm strains using four single nucleotide polymorphisms

Ashley E. Tessnow[1], Todd M. Gilligan[2], Eric Burkness[3], Caroline Placidi De Bortoli[4], Juan Luis Jurat-Fuentes[4], Patrick Porter[1,5], Danielle Sekula[1,6] and Gregory A. Sword[1]

[1] Department of Entomology, Texas A&M University, College Station, TX, United States of America
[2] Identification Technology Program, USDA-APHIS-PPQ-Science & Technology, Fort Collins, CO, United States of America
[3] Department of Entomology, University of Minnesota, St. Paul, MN, United States of America
[4] Department of Entomology and Plant Pathology, University of Tennessee, Knoxville, TN, United States of America
[5] Texas A&M AgriLife Research & Extension Center, Lubbock, TX, United States of America
[6] Texas A&M AgriLife Research & Extension Center, Weslaco, TX, United States of America

Corresponding author
Ashley E. Tessnow,
atessnow@tamu.edu

## ABSTRACT

The fall armyworm, *Spodoptera frugiperda,* is a polyphagous global pest with a preference for gramineous crops such as corn, sorghum and pasture grasses. This species is comprised of two morphologically identical but genetically distinct host strains known as the corn and rice strains, which can complicate pest management approaches. Two molecular markers are commonly used to differentiate between strains, however, discordance between these markers can lead to inconclusive strain identification. Here, we used double digest restriction site associated DNA sequencing to identify diagnostic single nucleotide polymorphisms (SNPs) with alleles unique to each strain. We then used these strain-specific SNPs to develop four real-time PCR based TaqMan assays to rapidly and reliably differentiate between strains and interstrain hybrids. These assays provide a new tool for differentiating between strains in field-collected samples, facilitating future studies on strain population dynamics and interstrain hybridization rates. Understanding the basic ecology of *S. frugiperda* strains is necessary to inform future management strategies.

## INTRODUCTION

The fall armyworm, *Spodoptera frugiperda* (Lepidoptera: Noctuidae), is an important economic pest endemic to the Western Hemisphere. This species is highly polyphagous, feeding on more than 357 different host plants (*Montezano et al., 2018*), but it causes most economic damage to gramineous crops such as corn, sorghum, and pasture grasses (*Johnson, 1987*; *Luginbill, 1928*; *Pashley, 1988*). Although *S. frugiperda* is considered

sporadic due to its migratory behavior (*Hardke et al., 2015*), severe fall armyworm infestations can result in complete yield loss, making them a significant agricultural threat. As a result, integrative pest management (IPM) measures have been implemented to control this pest and keep populations below an economic threshold (*Prasanna et al., 2018*; *Sparks, 1986*; *Wiseman, 1985*). The fall armyworm recently invaded the Eastern Hemisphere and spread rapidly across most of Africa, Southeastern Asia and into Australia, making this species a global concern (*Goergen et al., 2016*; *Otim et al., 2018*; *Sharanabasappa et al., 2019*; *Tay et al., 2020*; *Zhang et al., 2019*).

One recently proposed method for suppressing fall armyworm populations in the field is through the release of genetically modified (GM) fall armyworm adults carrying a self-limiting female-lethal gene (*Rwomushana et al., 2018*; *Stokstad, 2020*). Because wild moths that mate with the introduced GM moths would only produce male offspring that perpetuate the female lethal gene, the number of females in the population should dramatically reduce over several generations. This technique has been effective at reducing the populations of multiple insect pests, including the yellow fever mosquito (*Aedes aegypti*), the Mediterranean fruit fly (*Ceratitis capitata*), and the diamondback moth (*Plutella xylostella*) (*Asadi et al., 2020*; *Carvalho et al., 2015*; *Harvey-Samuel et al., 2015*).

One factor that complicates fall armyworm IPM, especially the potential for releasing GM insects, is the existence of two morphologically identical yet genetically distinct sympatric strains. These strains were originally described as host associated by *Pashley (1986)*, with the corn-strain (C-strain) primarily associated with large-blade grasses such as corn and sorghum, and the rice-strain (R-strain) primarily associated with small-blade pasture grasses (*Pashley, 1988*; *Pashley, 1986*). Despite these associations, the host range of these strains largely overlaps, and both strains are frequently collected in the same traps, especially in corn and sorghum fields (*Groot et al., 2010*; *Machado et al., 2008*; *Meagher & Nagoshi, 2004*; *Nagoshi & Meagher, 2004*; *Prowell, McMichael & Silvain, 2004*). Assortative mating (*Schöfl et al., 2011*) and hybrid sterility (*Kost et al., 2016*) have contributed to maintain genetic divergence between *S. frugiperda* strains (*Gouin et al., 2017*) suggesting interstrain hybridization is limited and releasing GM moths may only control one of the two strain types. In order for GM fall armyworms or any other strain-specific management approach to be effective, a much better understanding of strain population dynamics and frequency of interstrain hybridization is needed.

No consistent morphological differences have been found to differentiate between C-strain and R-strain fall armyworms to date (*Nagoshi et al., 2020*). As a result, molecular markers are required to distinguish between the two strains. Two molecular markers are commonly used to identify the strains. The first relies on polymorphisms in the mitochondrial *cytochrome oxidase I* (*COI*) gene, as described by *Nagoshi et al. (2006)*. These polymorphisms create unique restriction enzyme cutting sites that can be analyzed using RFLP analysis. The second marker used targets a series of single nucleotide polymorphisms (SNPs) in the Z-linked *Triosephosphate isomerase* (*Tpi*) gene (*Nagoshi, 2010*). Unfortunately, disagreement in strain identification between these two markers is often observed (*Schlum et al., 2021*) and has been reported in up to 24% of field-collected moths (*Nagoshi, 2012*), leading to some uncertainty in correct strain assignment. This

discordance could be the result of occasional hybridization, a phenomenon that has been observed in the lab (*Dumas et al., 2015*; *Kost et al., 2016*; *Pashley & Martin, 1987*; *Whitford et al., 1988*) and in field samples (*Schlum et al., 2021*). However, the fact that these genetic markers have not become homogenous over time suggests the two strains remain genetically distinct despite occasional gene flow (*Schlum et al., 2021*).

Recent advances in sequencing and real-time PCR allow for the development of novel molecular diagnostic assays based on single nucleotide polymorphisms (SNPs). Fluorogenic real-time PCR-based TaqMan®assays, originally developed by *Livak et al. (1995)* are a rapid and effective method for differentiating between specific alleles at a known SNP locus. This technology has been successfully used previously in *S. frugiperda* to detect insecticide resistance alleles (*Banerjee et al., 2017*). In this study, we used ddRAD-Seq data to identify novel SNP markers with unique alleles characterizing the two fall armyworm strains. We then developed four new real-time PCR-based TaqMan assays to rapidly and reliably differentiate between strains. These assays were validated using fall armyworm samples of each strain and their interstrain hybrids that were collected across both the central and eastern US flyways (*Westbrook et al., 2016*). Strain assignment results were compared to those of the two commonly used markers for strain differentiation, *COI* and *Tpi.* The assays developed here provide a new tool for identifying fall armyworm strains that enhance our confidence in strain determination and will facilitate future studies on population dynamics and interstrain hybridization in this important pest.

## MATERIALS & METHODS

### TaqMan assay development
#### Sample collection
*Spodoptera frugiperda* moths were collected using universal moth traps baited with Scentry PSU 2-component lures (Scentry Biologicals, Billings, MT) and containing Hercon Vaportape. Each trap was placed around corn and sorghum fields at five locations across the central US, and samples were collected at multiple time points throughout the year (Table 1). During each sampling time, traps were checked daily until a minimum of 24 moths were captured. At sites in the Lower Rio Grande Valley, larvae were occasionally hand collected from nearby host plants. All sampled insects were immediately preserved in 95% ethanol and stored at 4 °C until shipment to Texas A&M University in College Station, TX. Upon arrival, all specimens were stored at −80 °C until DNA extraction. In total, DNA from 426 moths was sequenced across two years.

#### DNA extraction
Prior to DNA extraction, the thorax was isolated from each specimen and surface sterilized in 95% ethanol. Tissues were tapped dry, placed individually in 2 ml Eppendorf tubes, and then frozen in liquid nitrogen. Sterilized plastic pestles were used to macerate the frozen thorax tissue. DNA was extracted using the Qiagen Gentra Puregene Tissue Kit, following the manufacturer's protocol. The concentration of each DNA sample was measured on a Fluorometer (DeNovix) and all samples were diluted to a concentration of 50 ng/µl.

**Table 1 Sampling location and collection date for all moths sequenced for the initial assay development.** Both the number of individuals per predetermined strain mitochondrial haplotype (R- or C-) and number of total individuals from each collection (R- + C-) is provided.

| Location | Date | GPS Coordinates | # Sequenced | | |
| --- | --- | --- | --- | --- | --- |
| | | | R- | C- | Total |
| Lower Rio Grande Valley, TX | 3–15 March 2017 | 26.1556, −97.9618 & 26.2099, −97.5432 | 22 | 16 | 6 |
| | 16 November 2017 | 26.1556, −97.9618 | 12 | 0 | 12 |
| | 10–11 May 2018 | 26.0924, −97.8814 & 26.0869, −98.2601 | 22 | 15 | 7 |
| | 12–13 July 2018 | 26.1556, −97.9618 | 23 | 22 | 1 |
| | 11–12 December 2018 | 26.1556, −97.9618 | 18 | 1 | 17 |
| Corpus Christi, TX | 18–20 April 2017 | 27.7827, −97.5621 | 20 | 1 | 20 |
| | 28–30 September 2017 | 27.7827, −97.5621 | 19 | 3 | 16 |
| | 12–13 May 2018 | 27.7827, −97.5621 | 18 | 10 | 8 |
| | 10–11 July 2018 | 27.7827, −97.5621 | 14 | 13 | 1 |
| | 7–8 October 2018 | 27.7827, −97.5621 | 13 | 0 | 13 |
| College Station, TX | 25–26 May 2017 | 30.6206, −96.3617 | 22 | 10 | 13 |
| | 6–7 July 2017 | 30.6206, −96.3617 | 16 | 16 | 0 |
| | 23–27 October 2017 | 30.6206, −96.3617 | 13 | 0 | 12 |
| | 16–18 May 2018 | 30.6206, −96.3617 | 18 | 7 | 11 |
| | 28–29 June 2018 | 30.6206, −96.3617 | 15 | 10 | 5 |
| | 19–24 October 2018 | 30.6206, −96.3617 | 12 | 0 | 12 |
| Lubbock, TX | 24–31 May 2017 | 33.6912, −101.8259 | 15 | 15 | 0 |
| | 21–27 June 2017 | 33.6912, −101.8259 | 23 | 16 | 7 |
| | 21 September 2017 | 33.6912, −101.8259 | 22 | 13 | 9 |
| | 2 May 2018 | 33.6912, −101.8259 | 12 | 12 | 0 |
| | 12 June 2018 | 33.6912, −101.8259 | 12 | 11 | 1 |
| | 13 September 2018 | 33.6912, −101.8259 | 18 | 13 | 5 |
| Rosemount, MN | 12–14 September 2017 | 44.7069, −93.1068 | 28 | 18 | 8 |
| | 21 August 2018 | 44.7069, −93.1068 | 20 | 20 | 0 |

### Strain haplotype determination

Prior to sequencing, strain identifications were assigned to each DNA sample based on two known RFLPs in the *Cytochrome C Oxidase Subunit I* (*COI*) mitochondrial gene, to ensure that individuals of both the C- and R- strain were represented in our collection (*Levy, Garcia-Maruniak & Maruniak, 2002*; *Nagoshi et al., 2006*). Briefly, the primer pair JM-76/JM-77 was used to amplify a 568 bp fragment of COI (*Levy, Garcia-Maruniak & Maruniak, 2002*). And aliquot of the amplicon (4 µl) was then digested with 50 units of SacI (New England BioLabs) and 50 units of MspI (New England BioLabs). Reactions were incubated at 37 °C for 1 h, and the products were run on a 1.8% agarose gel. The amplified C-strain mtDNA is cut once by MspI and not by SacI, while the R-strain mtDNA shows the reciprocal pattern (*Nagoshi et al., 2006*). Based on the cutting patterns of both restriction enzymes, each individual was assigned as having either a C-strain or an R-strain

mitochondrial haplotype. After haplotype determination, DNA was stored at −20 °C until sequencing.

### DNA sequencing, SNP calling and filtering

Purified DNA samples were sent to Texas A&M AgriLife Genomics and Bioinformatics Services (TxGen) for quality control, library preparation, and double digest restriction-site associated DNA sequencing (ddRADseq) (*Peterson et al., 2012*). Prior to library prep, DNA was purified using the Agencourt AMPure XP purification system. Libraries were prepared by digesting the total genomic DNA with MseI and EcoRI restriction enzymes, and 300–500 bp fragments were size selected for sequencing. Each fragment was ligated to standard Illumina adapters, sequencing primers, and multiplexing indexes. All sequencing was conducted on the Illumina NovaSeq 6000 to yield 150 bp paired end reads. Sequence cluster identification, quality prefiltering, base calling and uncertainty assessment was then conducted using Illumina's NCS 1.0.2 and RFV1.0.2 software with default parameter settings.

After demultiplexing and quality analysis by FastQC, sequences were uploaded into the Texas A&M High Performance Research Computing 'Ada' cluster for bioinformatic analyses. All sequences are now available through the NCBI Sequence Read Archive bioproject under accession number PRJNA645462.

FastQ Screen v.0.14.0 with the BWA aligner was used to align raw reads to both the C-strain (https://bipaa.genouest.org/sp/spodoptera_frugiperda_pub/download/genome/corn/v3.1/sfru.mais.corrected.3.1.fa) and R-strain (https://bipaa.genouest.org/sp/spodoptera_frugiperda_pub/download/genome/rice/v1.0/Spodoptera_frugiperda_rice_1.0.fa) published *S. frugiperda* genomes (*Gouin et al., 2017*). Sequences that did not match uniquely to one or both genomes were removed to clear the remaining sequences of all potential microbial or host contaminant DNA. Forward and reverse reads were then matched using the repair function in BBMAP v.3.8.08 (*Chaisson & Tesler, 2012*).

Genomic loci containing SNPs were identified using the dDocent v.2.2.16 pipeline (*Puritz, Hollenbeck & Gold, 2014*). In brief, dDocent removed low quality bases using Trimmomatic, and then mapped reads to the published chromosome map for *S. frugiperda* (*Liu et al., 2019*, https://db.cngb.org/search/assembly/CNA0003276/) using BWA. The program FreeBayes then identified genomic loci containing SNPs and indels, and these variants were concatenated into a single VCF file. Our initial VCF file contained 441,437 variants.

Variants were filtered using VCFtools v.0.1.16 (*Danecek et al., 2011*). Specifically, all indels were removed and the remaining SNPs were filtered for a minimum PHRED score of 30. Only SNPs that were present in all individuals at a minimum of 3x coverage were kept in the final dataset. Finally, the dDocent_filters script (https://github.com/jpuritz/dDocent/blob/master/scripts/dDocent_filters) was run to complete SNP filtering. After filtering, the VCF file was manually examined and 236 SNPs did not map to a specific chromosome but rather to an 'unplaced_scaffold.' These unmapped SNPs were removed, leaving 5,439 mapped SNPs in the final dataset.

### TaqMan assay development

To identify strain specific SNP loci, fixation indices ($F_{st}$) between individuals identified as having C- strain or R- strain mtDNA were calculated for each of the 5,439 mapped SNPs using R v.3.6.2/genepop (*Rousset, 2008*). All SNPs with $F_{st}$ values greater than 0.7 were identified for further analysis. Each of these SNPs were mapped to the *Liu et al. (2019)* chromosome map in Geneious v.11.0.2, and 250 bp upstream and downstream of each SNP was extracted (501 bp region total). If any of the other 5,439 identified SNPs were present in this 501 bp region, the reference nucleotide was denoted as 'N'. In several cases, divergent SNPs were in close proximity to one another in the genome.

All extracted sequences containing divergent SNPs were manually examined in Geneious to assess the variability in each region and the possibility of designing high quality primer and hydrolysis probe sequences. Of the 41 initial SNPs with $F_{st}$ values >0.7, four were selected to test for TaqMan real-time PCR assay development, referred to here as SNP A, B, C, and D. These SNPs were selected due to their spacing in the genome and their proximity to other known polymorphisms that could have interfered with primer and probe binding.

The 501 bp sequence containing each divergent SNP was uploaded to the Custom TaqMan Assay Design Tool offered through ThermoFisher Scientific (https://www.thermofisher.com/order/custom-genomic-products/tools/genotyping/). This program identifies the optimal primer and probe sequences for real-time PCR based SNP genotyping. In order to increase the melting temperature (Tm) of each hydrolysis probe while maintaining the short length, each probe was designed with a minor groove binder (MGB) moiety at the 3′end. Each probe contained either a FAM or VIC fluorescent reporter dye on the 5′end and a non-fluorescent quencher (NFQ) on the 3′end. Probes designed to bind to the R-strain and C-strain SNPs were bound to a FAM or VIC fluorophore, respectively. Primer and probe sequences are provided in Table 2. The context sequence and ThermoFisher assay IDs that can be used to purchase these custom assays are listed in Table S1. The development of these real-time PCR assays followed MIQE guidelines (*Bustin et al., 2009*).

## TaqMan assay validation
### Sample collection

We conducted three rounds of assay validation using (1) previously sequenced samples of known genotypes, (2) samples of unknown strain from the central US flyway, and (3) samples of unknown strain collected from the eastern US flyway. In the initial validation, 20 individuals that had previously been sequenced as part of the assay development were selected. These included 8 samples with the C-strain genotype, 8 with the R-strain genotype, and 4 samples that were identified as interstrain hybrids. In the second round of validation, DNA was extracted from 48 moths that were collected across the central flyway between 2017 and 2020. These moths had not previously been sequenced and the strain genotype was unknown. For the final validation, a collection of DNA from 44 moths collected between 2012 and 2017 at several locations in the eastern US was used. The collection information of all moths that were used in assay validation can be found in Table 3. During assay development all individuals assessed were males. However, if these assays are to be

**Table 2  Design information for all diagnostic assays included in this study.** The primer and probe sequences, chromosomal positions (*Liu et al., 2019*), alleles associated with the C- and R-strain, and the expected length of the PCR product are included for each TaqMan real-time PCR assay. Additionally, the primer sequences and expected PCR product length are listed for both the *COI* and *Tpi* diagnostic assays.

| SNP | Chrom | Position (bp) | PCR Product | C/R allele | Description | Sequence |
|---|---|---|---|---|---|---|
| SNP A | 1/Z | 14,104,488 | 62 bp | A/G | Forward Primer | 5′-GCAAGTGCAATTTTCCCATCTGATG |
|  |  |  |  |  | Reverse Primer | 5′-CAAGCCGTTCGCGGTTAG |
|  |  |  |  |  | FAM Probe Sequence | 5′-FAM-AGACCAAAAGGACTCAT-MGB-NFQ |
|  |  |  |  |  | VIC Probe Sequence | 5′-VIC-CTAGACCAAAAAGACTCAT-MGB-NFQ |
| SNP B | 1/Z | 4,933,322 | 113 bp | G/C | Forward Primer | 5′-GGGAACTCATATACTAAAATCGGAAAAACCT |
|  |  |  |  |  | Reverse Primer | 5′-ACACTCGCATTATTTGTGTGCAATT |
|  |  |  |  |  | FAM Probe Sequence | 5′-FAM-CCGCAGTAGCGTATGT-MGB-NFQ |
|  |  |  |  |  | VIC Probe Sequence | 5′-VIC-TCCGCAGTACCGTATGT-MGB-NFQ |
| SNP C | 1/Z | 4,683,787 | 57 bp | C/T | Forward Primer | 5′-TGACAGCATTGATGTGCTGGAT |
|  |  |  |  |  | Reverse Primer | 5′-CGCCGGAGCGTTACAGA |
|  |  |  |  |  | FAM Probe Sequence | 5′-FAM-CGCTACCAAAGCCAG-MGB-NFQ |
|  |  |  |  |  | VIC Probe Sequence | 5′-VIC-CGCTACCAGAGCCAG-MGB-NFQ |
| SNP D | 16 | 14,134,047 | 70 bp | C/G | Forward Primer | 5′-TGAGTGCCAACAGCTATCTTCTG |
|  |  |  |  |  | Reverse Primer | 5′-GCAGTCCATTACAGCTGGTGAA |
|  |  |  |  |  | FAM Probe Sequence | 5′-FAM-AGCTCATGTCCTACTCC-MGB-NFQ |
|  |  |  |  |  | VIC Probe Sequence | 5′-VIC-AGCTCATGTCGTACTCC-MGB-NFQ |
| *COI*[a] | mtDNA | - | 568 bp | - | *JM-76* Forward Primer | 5′-GAGCTGAATTAGG (G/A)ACTCCAGG |
|  |  |  |  |  | *JM-77* Reverse Primer | 5′-ATCACCTCC(A/T)CCTGCAGGATC |
| *Tpi*[b] | 1/Z | − | ∼600 bp | − | *Tpi-632* Forward Primer | 5′-GGTTGCCCATGCTCTTGAGTCCGGACTGAAGG |
|  |  |  |  |  | *Tpi-1195* Reverse Primer | 5′-AGTCACTGACCCACCATACTG |

**Notes.**
[a]Primers were originally developed by *Levy, Garcia-Maruniak & Maruniak (2002)* and made available by *Nagoshi et al. (2006)*.
[b]Primers were originally developed and published by *Nagoshi, Meagher & Hay-Roe (2012)*.

applied to female moths, it is important to avoid abdominal tissue as DNA within a mated female's spermatheca could interfere with strain assignment.

### *Strain determination using COI and Tpi*

Prior to determining the genotype of each individual at our four SNP loci, we assessed the strain of all samples at previously described *COI* and *Tpi* markers using RFLP analysis. As descibed above, a 568 bp fragment of the *COI* gene was amplified and then digested with the restriction enzymes MspI and SacI. The product was run on a 1.8% agarose gel to assess the restriction enzyme cut sites. Because *COI* is maternally inherited, hybrids could not be detected using this method (Fig. 1).

To identify strain using the *Tpi* marker, an approximately 600 bp fragment of the fall armyworm *Tpi* gene was amplified using the primer pair *Tpi-632F/Tpi-1195R* described by *Nagoshi (2012)*. In this reaction, initial denaturation occurred for 1 min at 94 °C, followed by 32 cycles of the following protocol: 92 °C for 30 s, 57 °C for 30 s, 72 °C for 1 min. The reaction was then held at 72 °C for three minutes. An aliquot (2 μl) of the PCR product was then digested with 50 units of MspI (New England BioLabs) following the manufacturer's protocol. This reaction was held at 37 °C for 1hr and then the resulting product was run

**Table 3  Collection information for all samples used to validate the four TaqMan real-time PCR assays.**

|  | Location | Date | # Samples | Strain if known |
|---|---|---|---|---|
| **Validation 1**: Known samples, sequences used in assay creation | Weslaco, TX | 11 December 2018 | 3 | R |
|  | Corpus Christi, TX | 13 May 2018 | 1 | Hybrid |
|  | Corpus Christi, TX | 10 July 2018 | 2 | C |
|  | Corpus Christi, TX | 19 October 2018 | 2 | R |
|  | College Station, TX | 24 October 2017 | 1 | Hybrid |
|  | College Station, TX | 18 May 2018 | 3 | R |
|  | College Station, TX | 28 June 2018 | 2 | C |
|  | Lubbock, TX | 2 May 2018 | 2 | C |
|  | Lubbock, TX | 12 June 2018 | 1 | Hybrid |
|  | Rosemount, MN | 12 September 2017 | 1 | Hybrid |
|  | Rosemount, MN | 21 August 2018 | 2 | C |
| **Validation 2**: Central Population Unknowns | College Station, TX | 20 October 2018 | 12 | ? |
|  | College Station, TX | 13 June 2020 | 24 | ? |
|  | Lubbock, TX | 2 May 2018 | 6 | ? |
|  | Rosemount, MN | 21 August 2018 | 6 | ? |
| **Validation 3**: Eastern Population Unknowns | Collier, FL | 15–24 February 2012 | 9 | ? |
|  | Tifton, GA | 7 August 2014 | 10 | ? |
|  | Jarretsville, MD | 3–30 August 2017 | 9 | ? |
|  | Charleston, SC | 30 June 2017 | 10 | ? |
|  | Roper, NC | 16 November 2017 | 6 | ? |

on a 1.8% agarose gel. The resulting banding patterns for C-strain, R-strain and hybrid individuals after RFLP analysis of both *COI* and the *Tpi* gene are shown in Fig. 1.

### Strain determination using TaqMan real-time PCR assays

Real-time PCR assays were conducted as 10 µl (final volume in TE buffer) reactions in 384 well plates. Each reaction contained 1 µl of template DNA diluted to 20 ng/µl, 5 µl of TaqMan Genotyping Master Mix (Applied Biosystems), 0.5 µl of 40×Custom ThermoFisher TaqMan assay containing the primers and hydrolysis probes (assay information listed in Table 2), 0.05 µl of Precision Blue Real-Time PCR Dye (BioRad). Non-template controls included 1 µl of TE buffer instead of DNA, and all reactions were conducted in duplicate. The real-time PCR program began by holding samples at 95 °C for 10 min, followed by 40 cycles oscillating between 95 °C for 15 s to 60 °C for 1 min. After being held at 60 °C for 1 min, fluorescence was recorded across all four channels of a CFX384 Touch Real-Time PCR Detection System (BioRad). All samples were assessed individually at each of the 4 diagnostic SNP loci (SNP A, B, C, and D).

All real-time PCR data were input into the CFX Maestro software (BioRad) to determine the quantification cycle (Cq) values (number of cycles for fluorescence detection above background threshold) for both FAM and VIC in each reaction. The CFX Maestro software identified each sample as either homozygous for the R-strain allele (primarily FAM fluorophore detected), homozygous for the C-strain allele (primarily VIC fluorophore
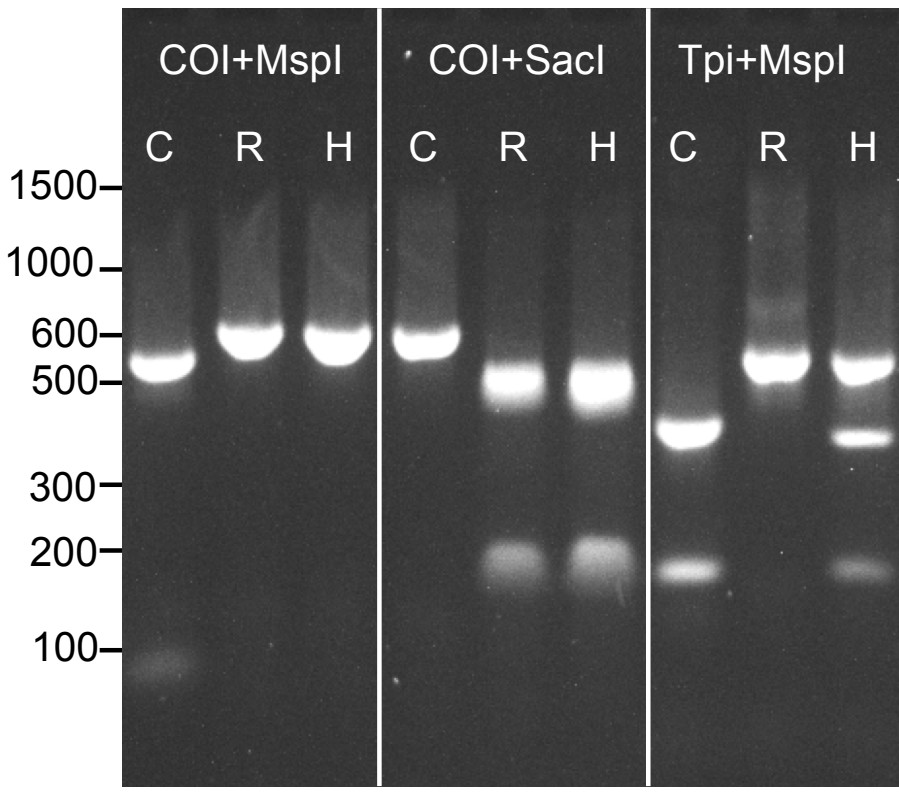

**Figure 1  Example RFLP analysis of known C-strain (C), R-strain (R), and interstrain hybrid (H) fall armyworms.** Amplicons for *COI* or *Tpi* were digested with MspI and SacI as indicated in the figure. Since mtDNA is maternally inherited, the hybrid could not be distinguished from the R-strain using *COI* alone.

detected), or heterozygous at the locus of interest (both fluorophores detected at similar levels). If both the FAM and VIC fluorophores were detected in a reaction, the difference between the Cq values of each fluorophore was calculated for each individual ($\Delta$Cq). The $\Delta$Cq across all homozygous individuals was then averaged for each allele and is presented as the $\Delta$Cq within each strain.

A consensus strain determination was made for all unknown individuals. This consensus was reached if a minimum of 5 out of the 6 diagnostic assays (4 real time PCR + 2 RFLP analyses) agreed on the strain call. The number of individuals that matched the consensus sequence was calculated for each assay. A chi-square test in JMP 14.0.1 (SAS Institute Inc,. Cary, NC, USA) was used to determine if there was a statistically significant difference in accuracy between the six diagnostic assays. In this case, accuracy was defined as the proportion of individuals matching the consensus strain assignment.

### Real-time PCR assay sensitivity analysis

The sensitivity of all real-time PCR assays was assessed using serial dilutions of DNA extracted from 3 R-strain and 3 C-strain fall armyworm moths. The samples selected for sensitivity analysis were all previously sequenced, and thus the allele present at each of the SNP sites was known. The DNA concentration was measured using a Fluorometer

(DeNovix) and each sample was then diluted to 100 ng/μl. Six serial dilutions were prepared by adding 1 μl of DNA to 9 μl of nuclease free water resulting in DNA concentrations of: 100, 10, 1, 0.1, 0.01, 0.001, 0.0001 ng/μl. The resulting Cq values of two duplicate runs were averaged for each SNP. No assay was able to detect fluorescence at 0.0001 ng/μl, so this concentration was removed from further analysis. The average Cq values across biological replicates were plotted against the log transformed DNA concentration. A linear regression was then fit in JMP® Pro 14.0.1 (SAS Institute Inc., Cary, NC) to determine the slope, y-intercept, and correlation ($R^2$) between DNA concentration and assay sensitivity

# RESULTS

## Diagnostic SNP identification

Of the 426 moths collected, 241 were identified as the C-strain and 185 were identified as the R-strain using known polymorphisms in the *COI* mitochondrial gene. After filtering the sequence data of these moths, 5,439 high quality biallelic SNPs were identified, 41 of which had Fst values higher than 0.7 between the two strains. This indicated that these SNP markers may exhibit a significant level of strain identity. Virtually all diagnostic SNPs identified were located on the Z-chromosome. Of the 41 initial SNPs with $F_{st}$ values > 0.7, four were selected to test for TaqMan real-time PCR assay development, referred to here as SNP A, B, C, and D. These SNPs were selected due to their spacing in the genome, their proximity to other known polymorphisms, and their Fst values. SNP A, B, and C are located on the Z-chromosome and had Fst values of ranging from 0.77 to 0.79 and SNP D was located on chromosome 16 that had an Fst value of 0.74. BLAST results suggest that these four SNPs do not result in changes to the amino acid sequence of any protein.

## TaqMan assay validation for four SNPs

In the first round of validation using samples of known genotypes, all four SNP-based TaqMan assays were able to accurately differentiate between C-strain (VIC fluorophore) and R-strain individuals (FAM fluorophore). Additionally, the SNP A and SNP D assays accurately identified all five-hybrid individuals as heterozygous at the SNP markers. Two of the hybrid individuals were identified as homozygous for the C-strain allele at SNP marker B and C. Upon further examination of our original sequencing data, we found that these individuals were indeed homozygous for the C-strain allele at these loci, indicating our assays correctly identified the individual genotype at the locus of interest, but hybrid individuals may not always be heterozygous at these loci.

In our final two validations using individuals of unknown genotypes from both the eastern and central US flyways, our assays detected a relatively even mix of C- and R-strain, with 42 individuals being consistently identified as the R-strain, and 50 individuals being consistently identified as the C-strain in at least 3 out of our 4 SNP assays.

The average Cq value for both fluorophores across all four SNP assays ranged from 22.91 to 23.93 in homozygous individuals. In most reactions, some level of fluorescence was detected for both FAM and VIC fluorophores by the end of cycle 40. In order to determine the number of cycles the target allele was detected before the off-target allele, we calculated the difference in the Cq value for each fluorophore within each strain. For

**Table 4  Mean ΔCq within each strain for reactions in which both fluorophores were detected.** This indicates the number of cycles after the initial strain specific fluorophore was detected until the secondary flourophore was detected above threshold. The FAM flourophore is associated with the R-strain whereas the VIC flourophore is associated with the C-strain.

|  | ΔCq ± SD within R-strain[a] | ΔCq ± SD within C-strain[b] |
|---|---|---|
| SNP A | Only FAM detected | 7.05 ± 0.89 |
| SNP B | 16.26 ± 1.26 | 17.12 ± 1.38 |
| SNP C | 10.32 ± 0.97 | Only VIC detected |
| SNP D | 3.67 ± 0.29 | 5.77 ± 0.45 |

Notes.
[a] In R-strain individuals the FAM fluorophore was detected first, so ΔCq was calculated as $Cq_{VIC}$-$Cq_{FAM}$.
[b] In C-strain individuals the VIC fluorophore was detected first, so ΔCq was calculated as $Cq_{FAM}$-$Cq_{VIC}$.

all homozygous R-strain individuals, the FAM fluorophore was detected after a lower number of cycles, therefore the ΔCq within the R-strain was calculated as $Cq_{VIC}$−$Cq_{FAM}$. For all homozygous C-strain individuals, the VIC fluorophore was detected after a lower number of cycles, therefore the ΔCq within the C-strain was calculated as $Cq_{FAM}$-$Cq_{VIC}$ (Table 4). The ΔCq for all heterozygote individuals across all SNP based TaqMan assays ranged from 0–2, indicating almost equal signal of both fluorophores. For SNP A, the VIC reporter dye was never detected above threshold for R-strain individuals, and for SNP C the FAM reporter dye was never detected above the threshold in C-strain individuals (Tables S2 & S4). For SNP B, some fluorescence was detected by both FAM and VIC regardless of the individual's allele, however the fluorescence of the off-target probe was always detected more than 15 cycles after the fluorescence of the target probe, which is sufficient to accurately identify the strain (Table S3). For SNP D, FAM and VIC fluorescence was detected in all individuals regardless of the allele present, and the fluorescence of the off-target probe was often detected only 3 cycles after fluorescence of the target probe (Table S5). This off-target probe binding could result in an overlap of fluorescence between the two reporter dyes, especially in assays with high DNA concentration. As a result, this marker may be less effective than the other three at differentiating between C- and R-strain individuals. Because of these limitations, when the sex of the individuals being genotyped is known to be male as is the case with most pheromone baited moth traps, we recommend using SNP A, B, and C for strain determination. Based on the results of the assay validations, instructions for implementing each of the four SNP based TaqMan assays are detailed in Figs. S1–S4. Additionally, Excel spreadsheet formulas (Microsoft®Excel Version 16.52) that can be used to determine strain from real-time PCR results are included in Table S7.

## Comparison between TaqMan assays, COI, and Tpi markers

In addition to the four SNP based real-time PCR diagnostic assays, we also determined the strain of fall armyworm samples collected in both the central and eastern US flyways (*Westbrook et al., 2016*), using two previously described diagnostic RFLP analyses (*COI* and *Tpi*). In 97.3% of individuals assessed, at least 5 of the 6 combined SNP and RFLP diagnostic assays were in agreement with the strain call (Table S6). In these cases, the consensus strain was determined for each individual. Three individuals did not show a consensus between assays. Two of these were previously sequenced individuals that were

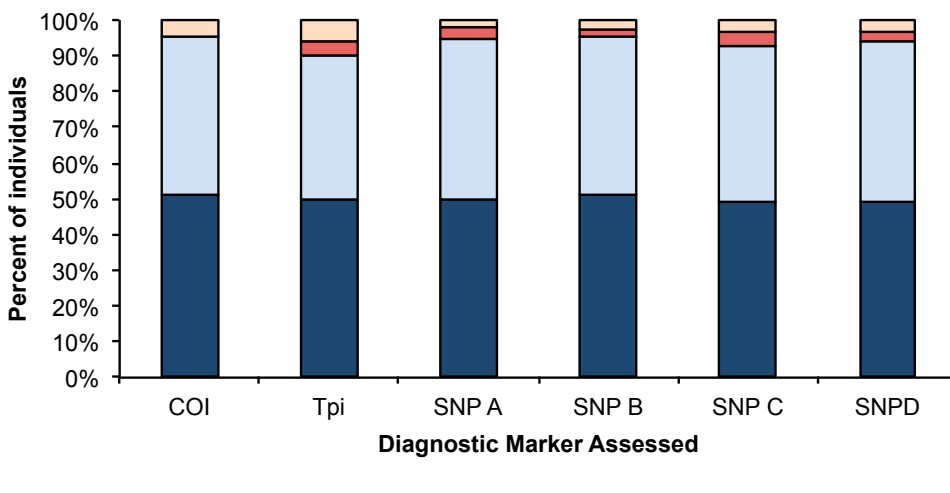

**Figure 2 Percent of individuals for each diagnostic assay that were in agreement with the consensus strain.** Each bar is split into individuals that were determined to be R-strain, C-strain, or hybrids. No significant difference was detected in accuracy between the six assays ($p = 0.07$).

known to be hybrids, and one was a non-sequenced individual from Collier (FL) that we suspect may also have been of hybrid descent. Given the unknown genotype of the individual from Collier, this sample was removed from assessments of assay accuracy.

For each assay, we determined the number of individuals that disagreed with the consensus strain call. The *Tpi* marker had the lowest accuracy with 7 individuals showing disagreement from the consensus strain, and SNP A had the highest consistency with 2 individuals showing disagreement with the consensus strain. We used a Pearson's Chi square test to determine if the proportion of accurate calls significantly differed amongst our six assays, and no differences were detected (Fig. 2, $X^2 = 17.14$, $p = 0.07$).

### Sensitivity analysis results

Serial dilutions of genomic DNA from known R- and C-strain individuals were used to assess the sensitivity of all four TaqMan real-time PCR assays to initial DNA concentration. Across all assays, no fluorescence was detected when <0.001 ng of DNA was added to the reaction. When more than 0.001 ng of DNA was added, there was a linear increase in the Cq of each reaction as DNA concentration decreased, with $R^2$ values ranging from 0.982–0.999 across all assays and fluorophores (Fig. 3). These results indicate that all four of our TaqMan real-time PCR assays are capable of detecting the allele present at each loci when using 0.001–100 ng of gDNA.

## DISCUSSION

We present four new diagnostic SNP markers to differentiate between the C- and R- fall armyworm strains. Prior to this study, the two most commonly used molecular markers to

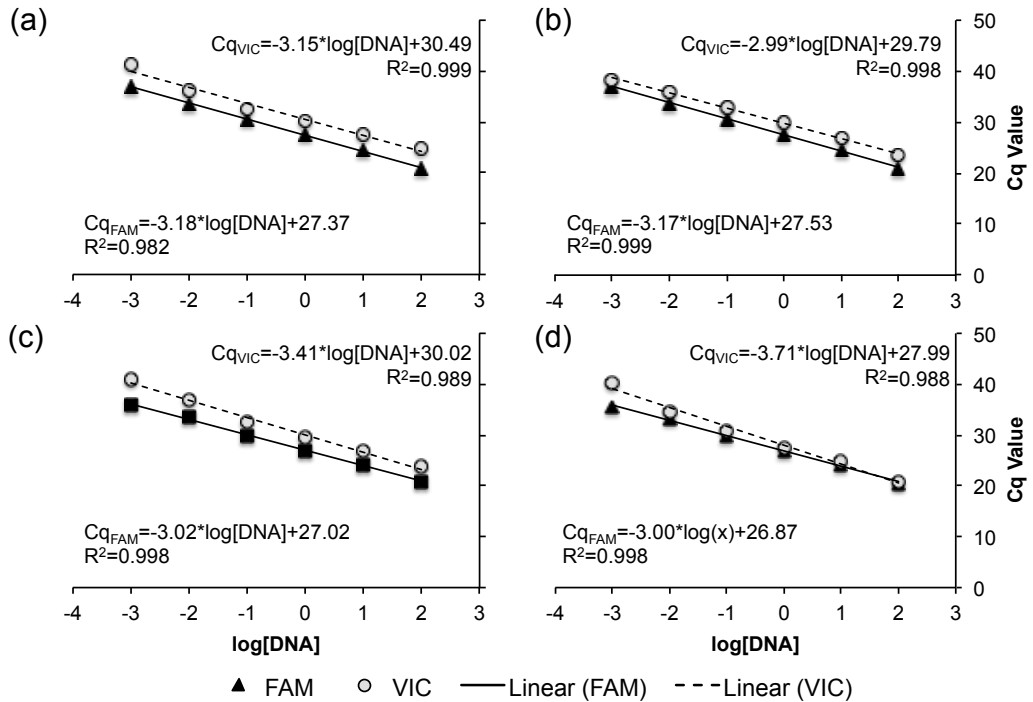

**Figure 3  Standard curve relating the DNA concentration within a reaction to the expression of both the VIC (dashed) and FAM fluorophore (solid line).** The standard curve for each TaqMan real-time PCR assays is demonstrated as a separate panel: (A) SNP A assay, (B) SNP B assay, (C) SNP C assay, and (D) SNP D assay. Corresponding slopes and $R^2$ values are reported.

discriminate these strains relied on polymorphisms in the mitochondrial gene *COI* or the Z-linked gene *Tpi* (*Nagoshi, 2010*; *Nagoshi et al., 2006*). However, discordance between these markers has been known to cause confusion in strain assignment. All four of the new SNP-based diagnostic assays were able to assign strain with equal accuracy when compared to the original *COI* and *Tpi* assays. Therefore, the addition of four new diagnostic markers will allow for increased confidence in the strain assignment. Furthermore, *COI* is maternally inherited and thus cannot identify hybrid heterozygotes, and *Tpi* is Z-linked and thus can only detect hybrids in males who have two copies of the Z chromosomes (ZZ). Three of our four diagnostic SNPs (SNPs A-C) were also located on the Z-chromosome and thus have the same limitations as *Tpi*, however, SNP D is located on chromosome 16, and thus could be used to identify hybrid heterozygotes of both males and females. This new diagnostic marker in the nuclear genome (chromosomes 2–31) could facilitate studies on interstrain hybridization in the field.

All four of the TaqMan assays presented here could effectively assign a fall armyworm strain when provided as little as 0.001 ng of DNA. This could be very useful for strain assignments when only a small amount of DNA is recovered. However, when 1 ng or less of DNA was present in the reaction, the Cq value increased above 30 cycles, potentially reducing the reliability of these assays. Therefore, if resources allow, we suggest that at least 10 ng of DNA be included in normal assay runs. The ΔCq value of the SNP D marker was

between 3-6 cycles indicating that both the target and non-target probes were binding to the DNA sequence containing the SNP. Although the target probe bound preferentially and was evident at least 3 cycles before the off-target probe, very high DNA concentrations may result in homozygous individuals being identified as hybrid heterozygotes due to a lack of probe specificity. When only 20 ng of DNA were added to the reaction this marker was as accurate at assigning strain as the rest. The probes for all other SNP markers were more specific, resulting in lower off-target probe fluorescence regardless of DNA concentration.

Across all individuals screened, at least five of the six diagnostic markers were in agreement (4 SNP assays + *COI* + *Tpi*). Therefore, to streamline the diagnostic process and reduce the number of reagents needed, only three diagnostic assays are required to confidently identify strain. Since all TaqMan assays require essentially the same reagents and have the same reaction conditions, they can be assessed simultaneously on the same real-time PCR plate. Thus, it would be most efficient to assess the strain of unknown fall armyworm samples using three of our SNP based diagnostic markers. Because of their higher reliability, we recommend that SNPs A-C be used when determining the strain of moths that are known to be males, as is often the case for most moth pheromone trap captures. Since male moths have two copies of the Z-chromosome, hybrid determination would still be possible and reliable under these conditions. However, if researchers are interested in determining the strain of females or individuals of unknown sex as is often the case with caterpillar collections, they may elect to use assays A, C, and D. Because females only have a single copy of the Z chromosome, the autosomal marker on chromosome 16 could then be used to more reliably identify hybrids. In the future, TaqMan assays could also be developed for the strain specific markers located in the *COI* and *Tpi* genes and run in concert with the SNP based markers described here. This would streamline the genotyping process across all six known diagnostic markers.

In the United States there are only two locations where fall armyworms are currently known to overwinter, south Texas and south Florida (*Luginbill, 1928*; *Nagoshi & Meagher, 2008*; *Sparks, 1979*; *Westbrook et al., 2016*), creating two populations in the US referred to as the central and eastern populations. These populations are commonly thought to be genetically distinct based on consistent differences in the ratios of mitochondrial haplotype that occur in each population (*Nagoshi et al., 2017*; *Nagoshi, Meagher & Hay-Roe, 2012*; *Nagoshi et al., 2008*). However, recent genomic studies using SNP data suggest that little genomic differentiation exists between the two geographic overwintering populations (*Schlum et al., 2021*). In this study, the identification of diagnostic SNPs was conducted using only data from the central population, but all four diagnostic markers were still capable of identifying the strain of insects across both the central and eastern populations. The efficacy of our diagnostic markers across both overwintering populations further supports the idea that some level of population panmixia occurs, despite geographic differences in the eastern and western overwintering locations.

Although the assays described here were validated using fall armyworm moths collected in the United States only, we expect that these assays would reliably differentiate between strains across the Western Hemisphere because several studies have shown that very little genetic differentiation exists between the central US and South and Central America

(*Schlum et al., 2021*; *Clark et al., 2007*; *Belay et al., 2012*). It is therefore likely that some level of population panmixia occurs throughout the hemisphere, resulting in similar genetic makeups across the large geographic range. More recently, fall armyworms were introduced and subsequently spread across Africa, Asia and Australia. Evidence suggests that the fall armyworm populations present in the Eastern Hemisphere descended from a population with a large proportion of C x R hybrids (*Nagoshi, 2019*). Although it is now thought that the majority of fall armyworms in the Eastern Hemisphere are most similar to the C-strain (*Nagoshi, 2019*), alleles typically associated with the R-strain may still be present in this invasive population due to their hybrid ancestry. As a result, it remains to be tested whether the SNP assays developed here are capable of differentiating strains in the newly invaded range.

As a novel management strategy, efforts are being made to reduce the fall armyworm numbers by introducing GM moths that contain a self-limiting female lethal gene. The success of this approach would be affected by dynamics of interstrain variation in fall armyworm population. Because the same mitochondrial haplotypes have been used to differentiate between strains over several decades, reproductive isolating barriers likely exist between strains as a result of either host use (*Pashley, 1986*), pheromone composition (*Groot et al., 2008*; *Lima & McNeil, 2009*), or nightly mating time (*Hänniger et al., 2017*; *Pashley, Hammond & Hardy, 1992*; *Schöfl, Heckel & Groot, 2009*). As a result, releasing GM moths of a single strain may only effectively reduce that strain, and could conceivably even increase the population size of the alternative strain through competitive release, for example. In order to better understand these dynamics, studies on the ecology and behavior of fall armyworm strains are of the utmost importance. Diagnostic assays that can rapidly and reliably differentiate between strains can provide the necessary tools to better understand these interstrain population dynamics.

## CONCLUSION

The fall armyworm species, *Spodoptera frugiperda,* is comprised of two morphologically identical strains that can only be distinguished using genetic markers. Here we identify four new diagnostic markers with unique strain-specific alleles. Using these markers we developed and validated Real-Time PCR assays to rapidly and reliably discriminate between strains in field-collected samples. These diagnostic tools facilitate future studies on strain ecology and population biology, allowing for better control of this economically important agricultural pest.

## ACKNOWLEDGEMENTS

We would like to thank Texas A&M Genomics and Bioinformatics (TxGEN) and the Texas A&M High Performance Research Computing (HPRC) for their services that made the sequencing and data analysis described in this study possible.

### Funding

This work was supported by the following: Agricultural and Food Research Initiative - Education and Workforce Development (award no. 2020-67034-31748) from the USDA National Institute of Food and Agriculture; Agricultural and Food Research Initiative (award no. 2021-67013-33566) from the USDA National Institute of Food and Agriculture; Agriculture and Food Research Initiative Foundational Program competitive grant (award no. 2018–67013-27820); Hatch Multistate NC-246 from the US Department of Agriculture National Institute of Food and Agriculture; and USDA-APHIS Cooperative Agreement (#AP19PPQS&T00C071). The funders had no role in study design, data collection and analysis, decision to publish, or preparation of the manuscript.

### Grant Disclosures

The following grant information was disclosed by the authors:
Agricultural and Food Research Initiative - Education and Workforce Development: 2020-67034-31748.
USDA National Institute of Food and Agriculture.
Agricultural and Food Research Initiative: 2021-67013-33566.
Agriculture and Food Research Initiative Foundational Program competitive: 2018–67013-27820.
USDA-APHIS Cooperative Agreement: #AP19PPQS&T00C071.

### Competing Interests

The authors declare there are no competing interests.

### Author Contributions

- Ashley E. Tessnow conceived and designed the experiments, performed the experiments, analyzed the data, prepared figures and/or tables, authored or reviewed drafts of the paper, and approved the final draft.
- Todd M. Gilligan conceived and designed the experiments, prepared figures and/or tables, authored or reviewed drafts of the paper, and approved the final draft.
- Eric Burkness, Caroline Placidi De Bortoli, Juan Luis Jurat-Fuentes, Patrick Porter and Danielle Sekula performed the experiments, authored or reviewed drafts of the paper, and approved the final draft.
- Gregory A. Sword conceived and designed the experiments, authored or reviewed drafts of the paper, and approved the final draft.

### Data Availability

All real-time PCR data from the four SNP based diagnostic assays is available in the Supplementary File.
The sequencing data used to identify diagnostic SNP loci is available at NCBI SRA: PRJNA645462.

## Supplemental Information

Supplemental information for this article can be found online at http://dx.doi.org/10.7717/peerj.12195#supplemental-information.

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
