# Peer review of "Novel real-time PCR based assays for differentiating fall armyworm strains using four single nucleotide polymorphisms"

_PeerJ, doi:10.7717/peerj.12195_

## Round 0.1 · original submission · Major Revisions

Dear Dr. Tessnow and colleagues:

Thanks for submitting your manuscript to PeerJ. I have now received two independent reviews of your work, and as you will see, the reviewers raised some concerns about the research. Despite this, these reviewers are optimistic about your work and the potential impact it will have on research studying fall armyworm diagnostics and ecology. Thus, I encourage you to revise your manuscript, accordingly, taking into account all of the concerns raised by both reviewers.

While the concerns of the reviewers are relatively minor, this is a major revision to ensure that the original reviewers have a chance to evaluate your responses to their concerns. There are many suggestions, which I am sure will greatly improve your manuscript once addressed.

Please consider discussing how the SNPs you identified apply to other geographic regions within fall armyworm distribution. Please also ensure that all of the relevant information for the figures and tables is provided.

Therefore, I am recommending that you revise your manuscript, accordingly, taking into account all of the issues raised by the reviewers. I do believe that your manuscript will be ready for publication once these issues are addressed.

Best,

-joe

·

Basic reporting

The writing is clear and unambiguous in most places. Sufficient background and citations are given. The structure conforms to PeerJ standards. Figures are relevant and well-described, except for some problems with supplemental figures described below. Most appropriate raw and summary data are available. I was able to access the BioProject PRJNA645462 containing the ddRADseq and download one of the Run Accession numbers, SRX8709398. According to lines 170-171 the reads were mapped to the published C and R genomes of Gouin et al. 2017; however there is more than one version of these genomes so the specific URLs should be given for the actual C and R genomes used. Also, the URL of the Liu et al (2019) genome should be given. I was able to access the TaqMan Assay Design Tool.

Experimental design

The manuscript describes original, primary research. The research question is well defined in a limited sense, but some ambiguity remains because the assumptions underlying the existence of fall armyworm strains are not defined (see below). Development and measurement of the TaqMan markers was rigorous and conducted to a high technical standard. Methods are described in sufficient detail for replication by others, subject to access to exactly the same genomes (see above). Replication by others would be improved by combining the criteria of FigureS1 through FigureS4 into a single Excel spreadsheet, with the decision rules given by formulas, and including this as additional Supplementary Information.

Validity of the findings

Although PeerJ does not assess impact and novelty, I provide some comments here. This work provides a novel method for rapidly screening large numbers of FAW individuals, once the method has been set up. It will have a measurable impact on the field because other workers will probably adopt the method. The markers appear to be robust, given the data included in the supplements; however applicability to other populations of FAW worldwide is unknown and can only be tested experimentally. The markers containing other non-diagnostic SNPs (B and D) are less robust than A and C, and future studies on other populations may also find SNPs in A and C, making them somewhat less robust.

Additional comments

1) The authors have invested more effort into developing these markers than any other previous study, and their methods will provide a blueprint for future marker development. However, whether the TaqMan markers provide a significant improvement over PCR-RFLP polymorphisms in the mitochondrial COI gene, and the 10 SNPs in the Z-linked gene TPI defined by Nagoshi, remains to be seen. Most researchers will still want to know the results of these "classical" tests--why not design TaqMan assays for them as well? Then a single platform could be used for both the old and new polymorphisms. Also, since three of the four new markers are Z-linked, they suffer from the same ambiguity as TPI. Why not develop one or two TaqMan markers for the W-chromosome? These might also be strain-specific, but even if they are not, they could be used to distinguish hemizygous C females from homozygous CC males, and hemizygous R females from homozygous RR males. This is necessary for a proper estimation of allele frequencies.

2) The genetic assumptions underlying the concept of "fall armyworm strains" are not explicitly laid out. What level of genetic differentiation is assumed, and does this vary by chromosomal position? Is it significant that three of the new markers are Z-linked, because sex-linked TPI was used beforehand?

3) The sampling seems biased toward corn-strain habitat. Would also sampling from rice fields have produced more robust markers?

4) It might be revealing to include information on the genomic context of the four markers. This can be done by BLASTing the probes against the annotated version of another FAW genome in GenBank, BioProject: PRJNA590312. SNP A occurs between 7,726,842 and 7,726,792 on the Z chromosome, in an intergenic region near XM_035575061, uncharacterized protein LOC118263201. SNP B occurs between 17,333,027 and 17,332,977, in the third position of the codon for arginine (CGG/CGC) of a protein-coding region of unknown function, SFRURICE_007604. SNP B does not change the amino acid sequence of this protein. The protein is a member of a large gene family that seems to be restricted to Spodoptera frugiperda. SNP C occurs in a noncoding region between 17,573,725 and 17,573,775, in the same general vicinity as SNP B. TPI occurs approximately between 14,025,953 and 14,026,704 on the Z chromosome, according to this coordinate system. SNP D occurs in the third position of a codon for serine (TCC/TCG) in gene XM_035580529 annotated as a solute carrier family 25 member; it does not change the amino acid sequence of the protein. The genomic context apparently has little relevance to the biological differences between the fall armyworm strains. What do the authors make of this?

5) Is discordance of the autosomal marker SNP D with the others, more or less significant than discordance of one of the sex-linked markers (A, B, C) with the others?

6) In Table 4 the caption seems to contradict the headings in the table. Are the deltaCq's reversed? Whether VIC or FAM always correspond to R or C is not made sufficiently clear in the text; it should be stated at least three or four times for clarity.

7) FigureS1 through FigureS4, change "stain" to "strain". For each box listing a test on the value of DeltaCq, state whether it is for VIC or FAM in each case, since it is not always obvious. "RT-PCR" usually means reverse-transcription PCR, but cDNA is not being used in this study. qPCR is the more common abbreviation for what the authors mean.

Reviewer 2 ·

Basic reporting

The manuscript provides a new molecular method to distinct genetically host strains of fall armyworm (Spodoptera frugiperda), a pest of global economic importance.

The authors bring a up-to-date tool and clearly state the practical relevance of their work regarding the population ecology and the importance for pest management strategy. For instance, reflected in the effectiveness using genetically modified insects release to control fall armyworm.

The manuscript is scientifically sound and well structured.

Experimental design

The material and methods are well described and reproducible. Nevertheless, I would like the authors to consider the improvements listed below:

1) Although the authors cite the nucleotide positions which SNP A, B, C and D are occurring, I would suggest adding a figure to illustrate the new position of SNPs (including which nucleotides are expected to be present in rice or corn strains). Are the SNPs analyzed located close to the regions (Tpi locus) analyzed by Nagoshi et al 2019 (Sci. Rep. 2019, 9, 8311) for fall armyworm populations collected in African countries?

2) Are the sequences (501 bp - line 196) used for TaqMan Assay design deposited in NCBI? What are the sizes of the PCR product formed for each SNP analyzed ?

3) Although it is clearly stated the tissue used for gDNA isolation in material and methods, I would like to suggest the authors to reinforce the importance of using a non-abdominal tissue in case moth specimens are used for the analysis. Considering that the target genes used in the TaqMan assay is a sex-linked gene and sperm present in the spermatheca could generate some result noise using such a sensitive method as qPCR.

4) Why did the authors considered using only Cq values instead of RFU?

6)The authors stated that the TaqMan assays developed have high sensitivity in detecting the different alleles at a concentration as low as 0.001 ng. However, if Cq values are used instead of final RFU as threshold for strain identification, I would question the reliability of the result using Cq values above 30, for instance NC-1, allele 1 in Table S3.

7) Line 126: were mixed sex used?

8) Line 145/242: add primers JM-76/JM-77; Tpi-632 F/Tpi-1195R in Table 2

9) Line 275: Include details of JMP 14 software used

Validity of the findings

The authors provide enough evidence to confirm the robustness of the new molecular tool proposed. Nevertheless, I ask the authorts to consider the following comment:

1) Although SNP D accurately identified the hybrids individuals, the authors also mentioned that “for SNP D, FAM and VIC fluorescence was detected in all individuals regardless of the allele present, and the fluorescence of the off-target probe was often detected only 3 cycles after fluorescence of the target probe” (lines 322-324). Later “All four of the new SNP- based diagnostic assays were equally if not more effective at assigning strain when compared to the original COI and Tpi assays (Lines 363-365). I question here rather only using SNPs A, B and C would not be more appropriate than suggesting the use of the four assays. The limitation of a Z-linked loci could be rather overcome if the sex of individuals would be known before hand.

Additional comments

Would the SNPs analyzed in this study support the strain identification of fall armyworm populations from other geographies outside the United States? I would encourage the authors to mention in the discussion the applicability of such method in a broader geographic range, or the corresponding limitations.

---

## Round 0.2 · accepted · Accept

Dear Dr. Tessnow and colleagues:

Thanks for revising your manuscript based on the concerns raised by the reviewers. I now believe that your manuscript is suitable for publication. Congratulations! I look forward to seeing this work in print, and I anticipate it being an important resource for groups studying fall armyworm diagnostics and ecology. Thanks again for choosing PeerJ to publish such important work.

Best,

-joe

Reviewer 2 ·

Basic reporting

The authors have covered all comments raised by reviewers and changes in the manuscript were made accordingly. The revised version of the manuscript is suitable for publication.

Experimental design

No additional comments.

Validity of the findings

No additional comments.